# Single-cell RNA sequencing reveals plasmid constrains bacterial population heterogeneity and identifies a non-conjugating subpopulation

Valentine Cyriaque [1,2] ✉, Rodrigo Ibarra-Chávez [1], Anna Kuchina[3,4,5], Georg Seelig [5,6], Joseph Nesme [1] & Jonas Stenløkke Madsen [1] ✉

Transcriptional heterogeneity in isogenic bacterial populations can play various roles in bacterial evolution, but its detection remains technically challenging. Here, we use microbial split-pool ligation transcriptomics to study the relationship between bacterial subpopulation formation and plasmid-host interactions at the single-cell level. We find that single-cell transcript abundances are influenced by bacterial growth state and plasmid carriage. Moreover, plasmid carriage constrains the formation of bacterial subpopulations. Plasmid genes, including those with core functions such as replication and maintenance, exhibit transcriptional heterogeneity associated with cell activity. Notably, we identify a cell subpopulation that does not transcribe conjugal plasmid transfer genes, which may help reduce plasmid burden on a subset of cells. Our study advances the understanding of plasmid-mediated subpopulation dynamics and provides insights into the plasmid-bacteria interplay.

To survive and reproduce, bacteria perform numerous tasks, including importing and processing nutrients and metabolites, cell division, defense/resistance, exogenous DNA acquisition, and many others. Yet, the number of tasks is limited by intracellular competition for relevant resources[1]. Transcriptional heterogeneity at the single-cell level, resulting in subpopulations[2], expands the number of simultaneous processes that can be performed at the population level. This can, for example, facilitate division of labor where complementary metabolic processes are carried out by different subpopulations[3]. Heterogeneity also enables bet-hedging strategies where mal-adapted phenotypes may become an asset in a changing environment[4]. The formation of subpopulations in isogenic populations[2] can thus increase the fitness of bacteria while preserving the genotype. Subpopulations can emerge through a number of processes and phenomena such as cell cycle progression, responsive switching in spatial gradients, epigenetic

determination[5], compartmentalization[6] and stochasticity during gene expression[1]. This results in transcriptionally and phenotypically distinct subpopulations[7] fluctuating over space and time, and has been shown to be important during utilization of different metabolic substrates[1] and survival of sudden physiological stress (e.g., via detoxification[3], competence[8] or persistence[9]).

Plasmids are extrachromosomal semi-autonomous mobile genetic elements (MGEs) that interact with their host's genome to varying degrees. Plasmids use host factors[10] such as nucleotides, tRNAs, amino acids, ribosomes[11] and thus take part in the intracellular competition for resources[2]. More crude interactions also transpire, facilitated via plasmid-encoded regulatory elements[9,12], toxin-antitoxin systems[13] or other defense and anti-defense systems[14]. The association of plasmids with a host can change its transcriptomic profile in a plasmid-host specific manner at the population level[15–17]. Plasmids can,

[1]Section of Microbiology, University of Copenhagen, Copenhagen, Denmark. [2]Proteomics and Microbiology Laboratory, Research Institute for Biosciences, UMONS, Mons, Belgium. [3]Institute for Systems Biology, Seattle, WA, USA. [4]Molecular Engineering and Sciences Institute, University of Washington, Seattle, WA, USA. [5]Department of Electrical and Computer Engineering, University of Washington, Seattle, WA, USA. [6]Paul G. Allen School for Computer Science & Engineering, University of Washington, Seattle, WA, USA. ✉e-mail: valentinecyriaque@gmail.com; jsmadsen@bio.ku.dk

for example, induce the use of alternative carbon sources or alter the metabolism of carbohydrates[16], amino acids and nucleotides[11]. Additionally, host respiratory activities[16] and regulatory pathways[16,17] can be directly influenced by plasmids. Over evolutionary time, compensatory mutations may reduce such genetic conflicts[11,18] between plasmids and their hosts. However, while the above illustrates that plasmids interact with host chromosomes, very little is currently known about plasmid-host interactions at the single-cell level and how these interactions may affect the subpopulation dynamics of their host.

Recently, several techniques successfully enabled single-cell transcriptomics in bacteria either by transcriptome imaging (Par-seq[19]), physically separating the cells with fluorescence-activated cell sorting (MATQ-seq[20,21]) or using random molecular-barcode ligation to concomitantly tag mRNAs stemming from the same cell (smRandom-seq[22], ProBac-seq[23], PETRI-seq[24], BacDrop[25], M3-seq[26], microSPLiT[7]). These latter approaches have made high-throughput single-cell RNA sequencing possible for a large number of prokaryotic cells in parallel.

Here, we successfully applied the microSPLiT (microbial split-pool ligation transcriptomics)[7] approach to assess how a conjugative broad-host range plasmid influenced the subpopulation dynamics of *Pseudomonas putida* KT2440. This bacterium is widely used as a model in studies of soil bacteria and is important for biotechnology[27]. *P. putida* KT2440 has also been considered for bioremediation (e.g.,[28]) and repeatedly used in studies of plasmid transfer in natural (e.g.,[29,30]) and synthetic communities[31,32]. We sought to investigate transcriptional interactions between a conjugative broad-host-range plasmid and the host genome at the single-cell level hypothesizing that; (i) *P. putida* would form distinct transcriptomic subpopulations despite growing in a homogeneous environment; (ii) single-cell transcriptomics would be successful at segregating plasmid free and carrier cells; (iii) the broad-host-range plasmid would interact distinctly with the host, in a sub-population dependent manner, and; (iv) the transcription level of plasmid-encoded genes would follow the growth status of its host to reduce its burden. In this work, bacteria grew with constant shaking in large bottles, making it unlikely that microenvironmental differences arose. Comparing bulk and single-cell transcriptomes, we observe clear differences at the individual-cell level leading to the formation of sub-populations. We find that the *P. putida* population is heterogeneous and

that the presence of a broad-host-range plasmid, which inflicts a population-wide burden, changes the subpopulation dynamics. The experimental design made it unlikely that transcriptional subpopulations identified were generated by arising mutations, as the cells only grow for a few generations and the mutation rate of *P. putida* is relatively low. Furthermore, our data demonstrate a differential transcript abundance of plasmid-encoded genes amongst bacterial subpopulations and that the transcription of operons, essential for conjugation, are absent in a subset of cells. These observations support that transcriptional heterogeneity has a determining role in the evolutionary and ecological success of plasmids, as seems to be the case for many bacteria.

## Results

### Plasmid affects growth but scantly chromosomal transcription
First, we sought to compare the growth of *P. putida* cultures in absence and presence of the broad-host range plasmid pKJK5 to evaluate any cost or benefit in the overall population. Notably, we detected a decrease in growth rate when carrying the plasmid at the start of the exponential phase (AUC $23.5 \pm 0.3$ vs. $24.9 \pm 0.4$, $p = 1.8 \times 10^{-9}$; r $0.76 \pm 0.03$ vs. $1.14 \pm 0.08$, $p = 5.12 \times 10^{-10}$) (Fig. 1A, Table S1). However, towards the end of the growth phase, no difference was observed. When growing plasmid-free and plasmid-carrier cells in coculture, in a competition experiment, the plasmid showed no significant fitness cost or advantage (w = 1,016∓0,009) after 16 hours of cocultivation (Table S2). First, unsupervised cluster analysis of the population-level (bulk) gene transcripts (excluding tRNA and rRNA) separated samples according to growth state (OD0.5 vs. OD1.5), and the presence/absence of the plasmid (Fig. S1). This separation, however, was mainly attributed to plasmid-encoded gene transcripts since, after excluding these, the separation between plasmid-free and carrier populations was no longer supported statistically (Fig. 1B), despite differential transcript abundance of a few chromosome-encoded genes.

### microSPLiT generates quality scRNA-seq data
Few studies have currently utilized high-throughput single-cell transcriptomics, like microSPLiT. Therefore, the acquired data were assessed by comparison to (i) population-level bulk transcriptomes (RNA-seq) and (ii) a blind microSPLiT replicate (referred to as the

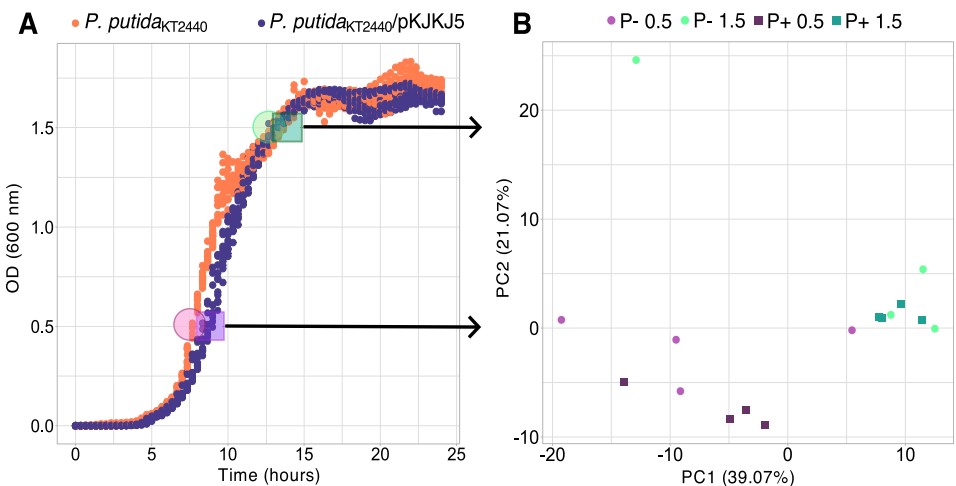

**Fig. 1 | Small growth effect of plasmid carriage with little influence on chromosomal gene transcription at the population level. A** Growth curve (OD = 600 nm) showing small differences in growth rates of *Pseudomonas putida* KT2440 with (blue) and without (orange) pKJK5 (Kruskal–Wallisχ² = 16.516; $p = 4.823e{-05}$) and area under the curve ($t = 10.045$; $p = 1.792e{-09}$; $n = 12$ independent experiments, see Table S1). Samples were taken at OD0.5 and OD1.5 for bulk transcriptomics or microSPLiT sample preparation (indicated by transparent colored circles following color code of **B**). **B** PCA multivariate analysis on

transcripts from chromosome-encoded genes at the population level (bulk; $n = 4$ independent experiments) according to the presence (P +) or absence (P−) of the plasmid and according to ODs (OD0.5 vs. OD1.5). The presence of the plasmid did not significantly discriminate between cell transcriptomes, as confirmed using 10,000 permutations in a two-sided PERmutational Multivariate ANalysis Of VAriance (PERMANOVA; $r^2_{plasmid} = 0.094$; $p_{plasmid} = 0.0922$; $r^2_{OD} = 0.253$; $p_{OD} = 0.001$; $r^2_{plasmid: OD} = 0.035$; $p_{plasmid: OD} = 0.7$). Source data and detailed statistical results are displayed in the Source Data file.

microSPLiT control) (Fig. S2). In contrast to the main microSPLiT experiment, the blind microSPLiT control was generated by mixing samples (plasmid presence/absence and growth state) before generating the microSPLiT libraries. The sum of mRNA transcripts per gene in the microSPLiT samples correlated well with those both from population-level (bulk) transcriptomes obtained from total mRNA extractions ($r = 0.793$) and the microSPLiT control ($r = 0.944$), showing that the single-cell data reflected population-level transcription. Importantly, this also indicates that microSPLIT did not introduce a systematic bias for specific transcripts. A total of 1599 single-cell transcriptomes (>85 transcripts/cell, Fig. S3) were recovered, characterized by transcripts from 4432 different genes (over 5698 mRNAs annotated in the genome) over all cells, with an average of 424.36 mRNA transcripts per cell. When performing microSPLiT, a trade-off between cell number (as defined during library preparation) and cell transcriptome coverage was seen: the microSPLiT control generated more single-cell transcriptomes (3165 cells) characterized by transcripts from 4248 genes across all cells with a lower average number of transcripts per cell (181.6 mRNA transcripts/cell) (Fig. S4). In bulk transcriptomes, 5632 different transcripts were obtained.

Among plasmid carrier cells, 91.4% displayed at least 1 plasmid-derived transcript at OD0.5 (average plasmid-derived transcript ratio = 1.24%) and 94.8% at OD1.5 (average plasmid-derived transcript ratio = 1.73%), suggesting that the plasmid is stably maintained in *P. putida*, as shown in previous studies[33,34]. The few percent of plasmid-encoded transcript free cells among plasmid carriers could result from a lower sequencing coverage per cell, segregational plasmid loss, or both.

For the following analyses, we filtered out transcripts that were present in less than 10% of cells for all growth conditions (P + 0.5, P + 1.5, P-0.5, P−1.5), resulting in 737 genes across 1486 cells (TableS S3, S4, Fig. S5). Single-cell transcriptomics showed the formation of distinct subpopulations constrained by plasmid carriage. Detailed transcript abundance per growth condition and by cluster can be found in Table S5. Unsupervised cluster analyses based on whole single-cell transcriptomes including both plasmid- and chromosome-encoded transcripts (reduced to the 10 top significant principal components, Fig. S6) generated 11 distinct clusters labeled 'W', projected on a uniform manifold approximation and projection (8 dimensions, UMAP) plot (Fig. 2A). Clusters segregated according to growth state and plasmid carriage (Fig. 2A, B). Single cell transcriptomes from samples grown to OD0.5 were mainly distributed in clusters W1-5, while cells from OD1.5 were distributed in clusters W6-11. Clusters W3-5 and W9-11 were dominated by plasmid-free cells, while plasmid carrier cells mainly distributed in clusters W1-2 and W6-8. (Fig. 2A–C). Chromosome-only single-cell transcriptomes formed 9 clusters (Ch1-9 clusters; Fig. S7, Table S5) showing transcriptional subpopulations of cells generally similar to W clusters. This overlay (Fig. S7C) suggests that the segregation between the transcriptome of plasmid-free and -carrier cells is not only due to plasmid-encoded transcripts, but also to changes in the transcription profile of the host chromosome (Fig. S8).

Genes differentially characterizing clusters W1-5 vs. W9-11 (i.e., OD0.5 vs. OD1.5; Fig. 3A, B) as identified with the Seurat function FindMarkers, were housekeeping genes including ribosomal proteins (*rps*, *rpl*, *rpm*), elongation factors (*fusA*, *tufB*), global regulators (*rpoC*), and malate:quinone oxidoreductase (*mqo*-II), whose transcript number was higher at OD0.5. Induced transcription of these genes is typical for early exponential growth[35] and Mqo-II, for example, is regulated by carbon sources and oxygen availability, which changed over time due to consumption[36]. At the end of the exponential phase (OD1.5), a high number of transcripts related to respiration in decreasing oxygen conditions characterized the samples: Differential transcribed genes included *cco*NOPQ-I, the cytochrome b component of the ubiquinol-cytochrome c reductase complex (*petB*), amino acid metabolisms (*arc*ABCD-1, *gdhB*), and flagellin (*fliC*). These observations were confirmed with population-level (bulk) transcriptomics comparing OD0.5

and OD1.5. A few genes discriminated between plasmid-free and carrier cells (Fig. 3C, D), such as those involved in rRNA maturation, protein translation and translocation (*rne*, *lnfC*, *secD*, *tig*) or the 3-hydroxydecanoyl-dehydratase (*fabA*) whose increased transcript number in plasmid carrier at OD0.5, may have consequence for the membrane fluidity. Indeed, at the outset of growth, flow cytometry analyses revealed a small change in size and cell texture (Fig. S9). These changes may facilitate conjugation and plasmid maintenance.

The few differentially transcribed chromosome-encoded genes identified at the population level (bulk transcriptomics), did not discriminate between plasmid-free and carrier cells. Looking at single-cell transcriptomes only, clusters W1, W4, W7, W10 and W11 ('Ribosome-rich subpopulations') were distinct from clusters W2, W3, W5, W6, W8, and W9 ('Ribosome-poor subpopulations') (Fig. 2A, D). Interestingly, the underlier of this separation did not seem to be related to OD or plasmid presence/absence. Instead, subpopulations W1, W4, W7, W10, and W11 were characterized by increased levels of transcripts encoding ribosomal proteins (*rpl*, *rpm* and *rps*). This suggests that subpopulations with distinct translation rates or cell cycle stages co-occurred regardless of the growth state. Furthermore, an increased rate of ribosomal gene transcription co-occurred with an increased abundance of transcripts of house-keeping genes, isoprenoid biosynthesis and, in plasmid carrier cells, to plasmid encoded genes (Fig. 2D).

At the beginning of the exponential phase (OD0.5), plasmid-free cells were evenly distributed among subpopulations W3, W4, and W5 (Fig. 2C) with the latter almost exclusively populated with plasmid-free cells. However, W5 cells were characterized by transcripts that also defined cells at OD1.5. Indeed, when mapping W5 transcriptomes with the transcriptomes from the microSPLiT control (Fig. S10), we see that those cells cluster with cells with OD1.5 transcription profiles (including clusters W6-9). The W5 cluster exhibits increased transcript abundance of genes involved in (i) glycolysis (*eda*), (ii) the *arginine* deiminase pathway and L-argnine metabolism (*arc*ABCD, *kau*B), lysine oxidative decarboxylation (*dav*B), (iii) glutamine synthase (PP-3148) and (iv) the oxidation of L-glutamylputrescine (*puu*B) (Fig. 2). This subpopulation of cells was also characterized by having many transcripts involved in motility (*fliC*), and respiration (*ccoP*). The chromosome single-cell transcriptome data suggest that cells in the W5 subpopulation shifted toward transcriptional patterns to an alternative source of energy, and nitrogen through arginine and glutamine metabolisms.

Notably, at the end of the exponential phase (OD1.5), subpopulations of cells displaying an increased number of ribosomal gene transcripts (W7, W10, and W11; Fig. 2D), also displayed more transcripts of efflux pumps (*opr*CQ), for nucleic acid and amino acid synthesis, degradation and import (*tkt*A, *cmk*, *put*P, *gch*V-I, PP-0496, *asp*A), regardless of the plasmid (Fig. 2D, "OD1.5 subpopulation").

## Plasmid-encoded genes are differentially transcribed among cells

The subpopulation dynamics at OD0.5 seemed to be driven by the presence of the plasmid, which may have led to the decreased growth rate of the plasmid carrier cells, perceived at the outset of growth (Fig. 1A). The burden of the plasmid may also have resulted in the observed decrease of the cluster coefficient in plasmid carrier cells (degree to which genes tend to cluster together; Table S6), in network graphs calculated from spearman correlations between genes of each single-cell at OD0.5 (>85 transcripts/cell; Fig. S11). Reduced co-occurrence correlation between chromosomal genes of plasmid carrier cells suggests that the plasmid interferes with chromosome transcriptional regulation. Furthermore, when calculating the Spearman correlation between plasmid and chromosomal genes (Fig. 4A), we see that plasmid encoded genes, especially *tra* genes, were transcribed concomitantly ($n > 30$, $p < 0.05$) with *mqo*-II and ribosomal protein transcripts (*rps*I, *rps*P and *rpm*I, *rpm*J genes). Additionally, *tra* gene

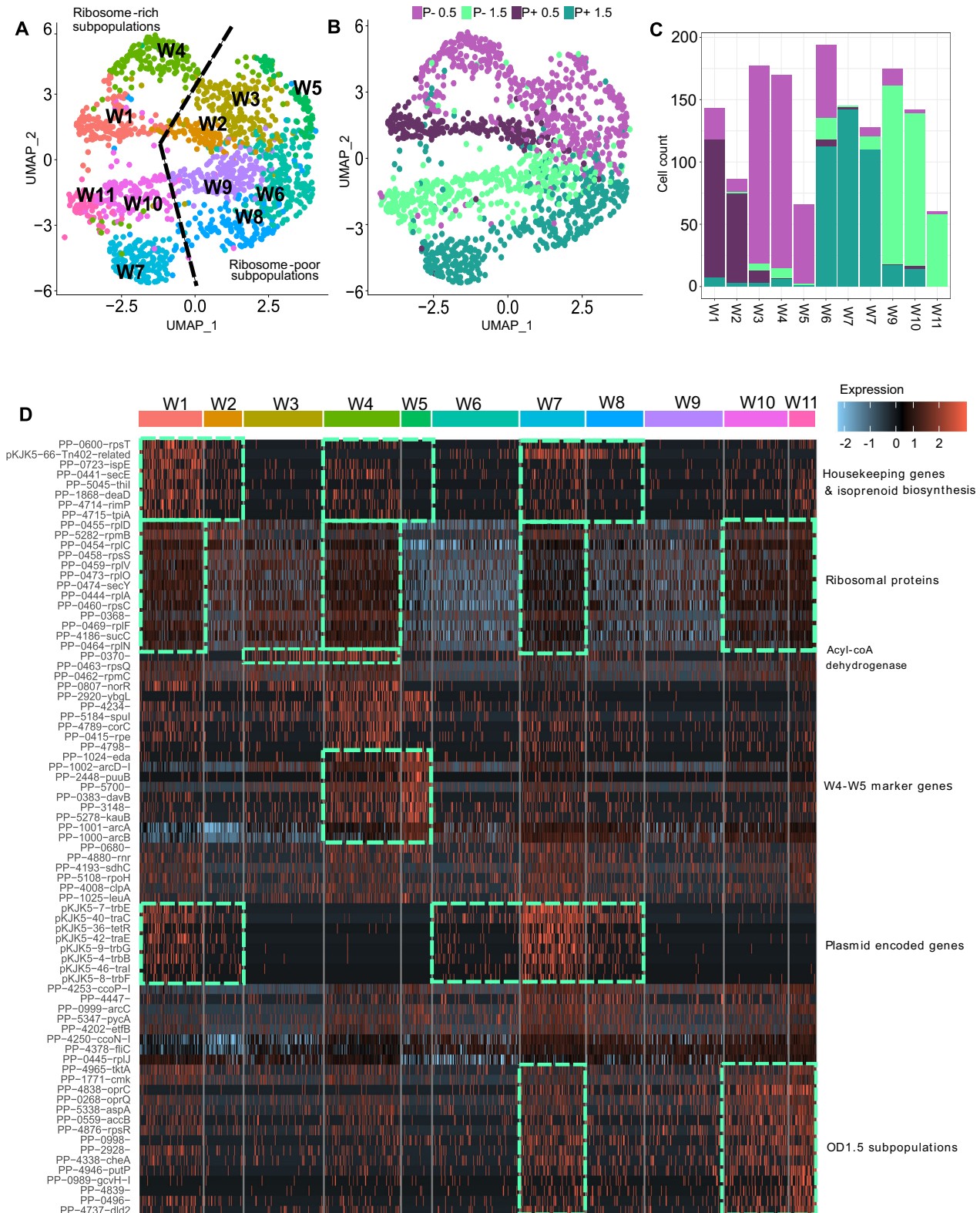

transcription exhibited significant positive correlations (rs > 0.6, n > 30, p < 0.05) with transcripts involved in LPS biosynthesis and membrane protein translocase and chaperones (*lpt*D, *rfb*AD, *tig*, PP-2304), DNA synthesis (*nrd*B, *pur*A), peptide transport, rRNA maturation, protein synthesis and folding (*dpp*A-II, *rnr*, *rbb*A, *fkl*B), respiration, energy production, oxidoreductases, oxidative stress (*cyo*AC, *fpr*I, *nuo*F, *atp*B, PP-2010, *sod*B) and carbon metabolism and coordination (*gpm*, *pyc*AB, PP-3443, *spo*T), especially after the exponential phase (OD1.5). Intriguingly,

these chromosomal gene transcripts also positively correlated with pKJK5-encoded Tn402-related gene transcription (Fig. 4A).

Transcription of the *tra*GFEDC operon specifically, which encodes genes involved in DNA replication and transfer, critical for conjugative plasmid transfer, was further investigated (Fig. 4). The P$_{traG}$ promoter of the *tra*GFEDC operon, was fused with the green fluorescent protein sfGFP (P$_{traG}$-*sfGFP*) on the pPROBE-NT vector[37] to enable quantification of single-cell expression levels by flow cytometry. Both microSPLiT

**Fig. 2 | Whole single-cell transcriptomes generally cluster in accordance with growth state and the presence/absence of the plasmid.** Subpopulation clustering identified by single-cell transcriptomics of *P. putida* carrying a plasmid (P +) or not (P−) at early (OD0.5) and late exponential growth (OD1.5) (*n* = 1 independent experiment; 1486 cells). **A** UMAP obtained by microSPLiT scRNA sequencing identified 11 whole transcriptome clusters (W1-W11). Clustering was performed with an integrative approach combining Euclidean distance-based K-nearest neighbor, refining the edge weights with the Jaccard similarity and the Louvain algorithm. Color of the dot indicate to which cluster the cell belongs as indicated on the graph (W1-11). **B** Same UMAP graph where color of the dot indicate the growth state (OD0.5 and OD1.5) and the presence of the plasmid (P +/P−). **C** Cell

transcriptomes from OD0.5 were mainly distributed in clusters W1-5, while cells from OD1.5 were mainly distributed in clusters W6-W11. Data are presented as absolute cell count. **D** Heatmap of normalized and centered-scaled transcript number per cell as displayed through the expression color code (normalization consist in a log10 transformation of transcript numbers * 10,000 divided by summed transcript number of the cell). Top colors represent the cell cluster (W1-11). The top 8 biomarker genes (*p* < 0.05) identified by a two-sided Wilcoxon signed-rank test are shown for each cluster (common biomarker to multiple groups where displayed only once). The *p*-value associated with each marker gene for each cluster (cluster of interest vs. all cells) can be found in Supplementary Dataset 1. Source data are provided as a Source Data file.

single cell transcriptomes and flow cytometry of the P$_{traG}$-*sfGFP* reporter showed, that *tra*GFEDC transcription was increased at OD1.5, as more cells in the population had a positive transcription signal (GFP fluorescence and *tra*GFEDC transcripts; Fig. 4B, C) and the level of transcription signal in individual cells was higher (fluorescence intensity and transcript number; Fig3D). Furthermore, an unsupervised cluster analysis was performed including only plasmid transcripts from plasmid-carrier cells (>85 transcripts/cell). Transcript numbers were normalized by dividing transcript counts by the summed numbers of plasmid-encoded transcripts and performing log+1 transformation (Fig. 5). Despite a low significance of the principal component, due to low transcript numbers, 6 clusters (P1-6) of plasmid transcriptomes were identified (Fig. 5A), grouping cells independently of growth state. Interestingly, the Tra2 region (summed *trb* genes encoding the mating pair formation including pilus assembly) was homogeneously expressed among plasmid carriers (Fig. 5C). Yet, the transcriptional regulator TrbA that targets P$_{trbB}$ and the Tra1 region, was present in a small number of cells that also had fewer transcripts of other *trb* genes (Fig. 5D). Also, the P4 plasmid cluster which represented about 12% of plasmid carriers, was distinguished by a lack of transcripts from the *tra*GFEDC operon (Fig. 5E, F). The occurrence of a subpopulation of cells not expressing the *tra*GFEDC was confirmed by the GFP-based reporter plasmid where P$_{traG}$ was inactive in about 13% of the cells (cells with no detectable GFP signal). Nonetheless, the P4 subpopulation did not appear to be inactive or consist of dead cells, as the numbers of *rps* and *rpl* ribosomal protein transcripts were equivalent to those of other plasmid-carrier cells (Fig. S13).

## Discussion

Here, we investigated transcript abundance at the single-cell level of *P. putida* with and without the conjugative broad-host-range IncP-1 plasmid pKJK5. With microSPLiT, we observed that *P. putida*, despite growing in a continuously homogenized environment, displayed transcriptional heterogeneity. This is in agreement with the few other studies using similar approaches[7,24,25] on bacteria belonging to different orders. Cells clustered into transcriptional subpopulations, displaying differential expression of ribosomal proteins and alternative amino acid and catabolic, energy-providing pathways. Generally subpopulations may arise due to microenvironmental differences, epigenetic determination, phase variation[2] and mutations[38]. The formation of subpopulations has the potential to improve the population-wide fitness of the genotype by facilitating bet-hedging or division of labor strategies. In this study, the subpopulation W5 of plasmid-free cells observed at OD0.5 appeared to undergo a diauxic shift and/or co-utilization of energy sources[39], which have been shown to occur through stochastic events leading to bi-phasic shifts within isogenic populations[40]. The use of an alternative energy source by plasmid-free cells may have led to the differences in overall growth rates observed between plasmid-free and carrier cells since, during early exponential growth, no plasmid carriers adopted the transcriptome profile of subpopulation W5. The stability of pKJK5 within a bacterial population is ensured by conjugation[33], genes ensuring stable inheritance (partitioning system *par*A and *inc*C; *klc* and *kle* genes[33]), a putative toxin-antitoxin system[41], and a replication system that enlarges the host

range of the plasmid, especially among proteobacteria[41]. However, its stability in a subpopulation of cells seems to also depend on the transcriptomic context. The aforementioned shift to an alternative energy-providing pathway, combined with the cost of plasmid carriage, may have induced a fitness loss, impeding *P. putida* from meeting the burden of plasmid maintenance[42], by reducing the number of formed subpopulations. The W5 subpopulation was characterized by a higher number of transcripts involved in arginine deiminase pathway leading to the expulsion of ornithine. The plasmid may thus have deprived the population of plasmid carriers of an alternative ATP/NH$_3^+$ source in addition to extracellular ornithine, which interestingly, was also shown to act as a public good for *Streptococcus* species[43,44].

When investigating correlations between plasmid- and chromosome-encoded transcript abundances among plasmid-encoded genes, we found that *tra* genes involved in DNA replication for transfer, were co-transcribed with housekeeping genes, which imply that the pKJK5 plasmid decreases its burden by aligning with its host's activity. Likewise, we show that the abundance of transcripts associated with the Tn-402-related transposon encoded by pKJK5 was also correlated to the same housekeeping genes. Similar observations were made in another single-cell transcriptome study[25] and we speculate that many MGEs may, at least in part, rely on, or synchronize with, the cell's transcriptional activity for the induction of transposition or horizontal mobilization, as observed with other transposable elements (e.g.,[45]), which could be advantageous for the fitness of the host and the success of the MGE itself. Lastly, we show with both microSPLiT and the P$_{traG}$-*sfGFP* promoter activity reporter, that a defined subpopulation of cells did not transcribe the *tra*GFEDC operon and thus did not engage in conjugation. Interestingly, this subpopulation did transcribe ribosomal proteins and did, therefore, not appear to be either inactive or dead. It may seem counterintuitive that the plasmid does not maximize its potential for horizontal transfer; however, decreasing the expression of the large conjugative machinery in a subpopulation of cells, and thus it's burden on the host cell, may represent a bet-hedging strategy by which the plasmid promotes its association with the genotype, in a specific subset of cells.

This study is among the first to discern the role of bacterial subpopulations in plasmid biology. Using microSPLiT, we illustrate that transcriptional heterogeneity among cells can indeed be central to plasmid-host genome interactions. Plasmid carriage impacted the subpopulation dynamics of the host, and the transcript abundance of plasmid-encoded genes was highly heterogeneous. Central plasmid functions such as replication and maintenance seem to fluctuate, probably in sync with the cell cycle, while transcript abundance of genes critical for conjugation was biphasic, resulting in a smaller subpopulation that likely does not engage in conjugation. Studying subpopulation dynamics holds great promise for the understanding of bacterial ecology and evolution and for biotechnological applications. An important perspective of the findings presented here is that we need to further understand single-cell heterogeneity in plasmid biology at a broader scale and in natural environments where microbes experience a large diversity of environmental factors, and face a large variety of MGEs, like plasmids, ICEs, phages, PICIs and transposons,

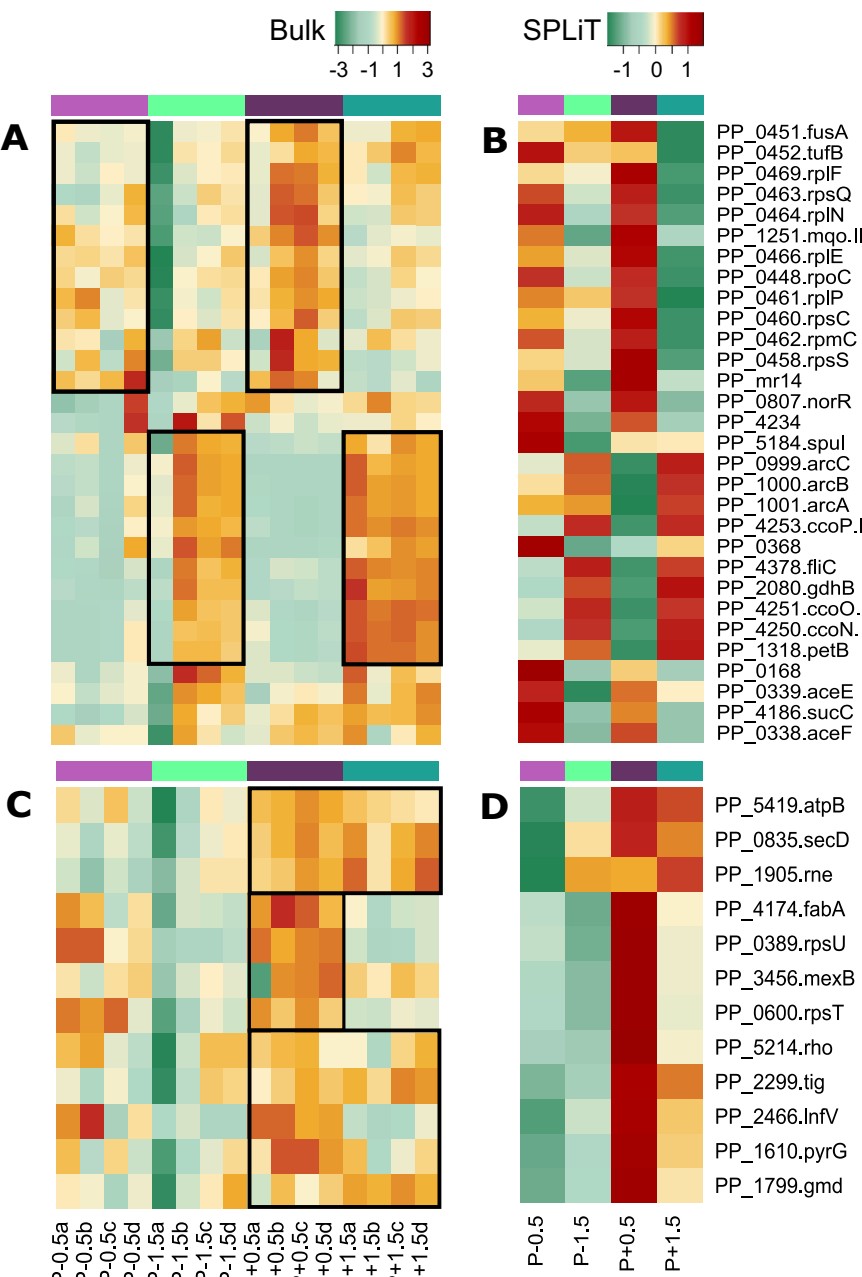

**Fig. 3 | A few genes discriminate between samples according to OD and the presence of the plasmid.** Heatmaps obtained from (**A**, **C**) the normalized population (bulk; *n* = 4 biological replicates) and (**B**, **D**) the summed normalized microSPLiT (scRNA seq; *n* = 1 independent experiment) normalized transcription rate of genes characterizing cells categorized by growth stage (OD0.5 or OD1.5) and plasmid presence (P−/P+) showing the 30 first marker genes (two-sided non-parametric Wilcoxon rank sum test; *p* < 0.01) obtained with Seurat separating (**A**, **B**) clusters "W1", "W2", "W3", "W4", "W5" vs. "W6","W7", "W8", "W9", "W10", "W11" mainly composed of OD0.5 and OD 1.5 cells, respectively; and (**C**, **D**) separating clusters "W1", "W2" vs."W6", "W7", "W8", and "W6","W7", "W8" vs. "W9","W10", "W11" mainly composed of plasmid carrier- or free- cells at each OD, respectively. Normalization was operated by dividing genes counts by the total count of the sample *10,000 as displayed on the color key. Source data are provided as a Source Data file.

enhancing genetic variability among populations (e.g.,[46,47]). Therefore, and in the context of the ongoing antibiotic resistance crisis, a deeper understanding of the interactions between plasmids and other MGEs with their host will be critical.

## Methods

### Bacterial strains, cultures, growth curve, and competition experiment

*Pseudomonas putida* KT2440 (ATCC™ 47054) growth was performed on Lysogeny-Broth medium (LB) at 30 °C and 250 RPM. *Pseudomonas putida* KT2440/pKJK5 was obtained by electroporating 1 µg of pKJK5

plasmid (private collection) into 50 µL of competent *P. putida* KT2440. Competency was achieved at an optical density (600 nm) of 0.8. A culture of 50 mL was washed twice with 50 mL of 10% ice-cold glycerol and finally resuspended in 800 µL of 10% ice-cold glycerol. Cells were aliquoted (50 µL) and electroporated in a 2 mm cuvette at a voltage of 2400 V, 1 pulse. For the following experiments, *P. putida* KT2440 and *P. putida* KT2440/pKJK5 were pre-cultured separately overnight in 5 mL of LB medium. *Pseudomonas putida* KT2440/pKJK5 was cultivated with tetracycline (50 µg/mL). For the growth curves (3 biological replicates, 4 technical replicates), pre-cultures were washed in LB medium and diluted 10⁴ times before distributing 200 µL in a 96-well plate

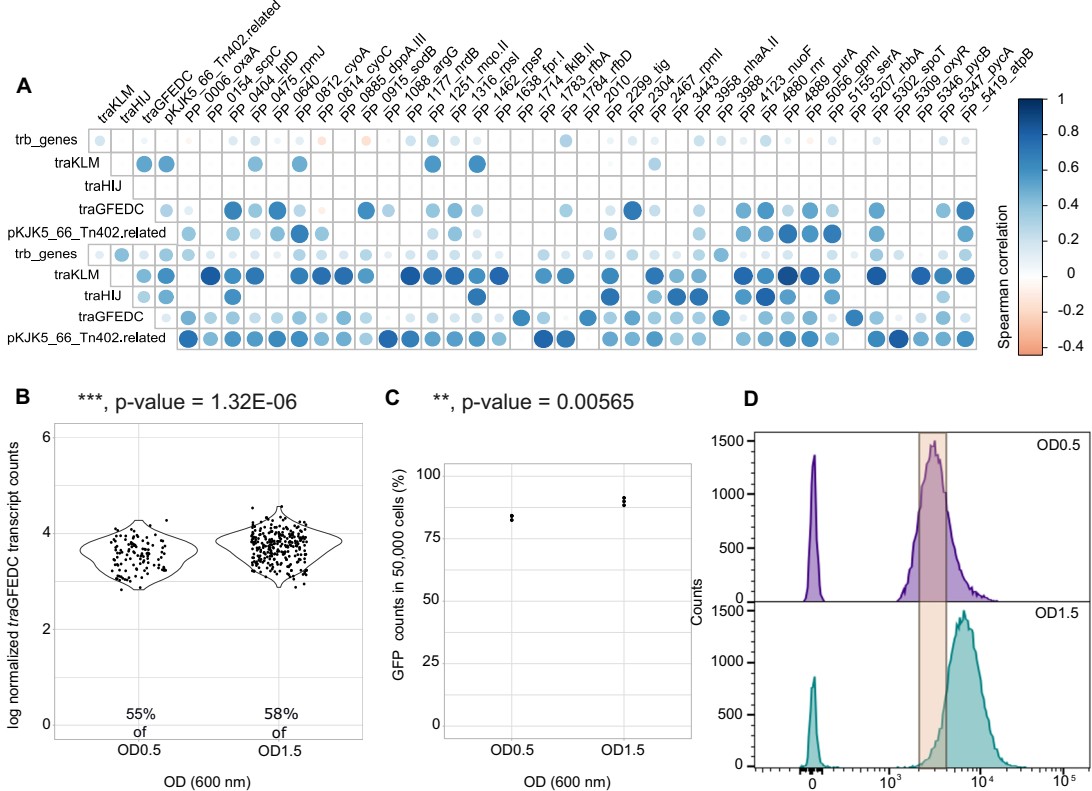

**Fig. 4 | Subpopulation transcriptional heterogeneity of the *traGFEDC* operon, which is essential for conjugation, validated by flow cytometry of the promoter-*sfGFP* fusion reporter. A** Spearman correlation matrix between plasmid genes with several chromosome-encoded genes. Significance was determined with a two-sided *t*-test by the rcorr-5.1-2 R package (*n* > 30 cells, *p* < 0.05). The size of dot is proportional to the correlation (**B**) Normalized summed transcript counts of *tra*GFEDC genes from individual cells on a violin plot according to OD (*n* = 1 independent experiment; *n* = 384 *tra*GFEDC +/P+ cells). **C** Percentage (mean ± SD; *n* = 3 independent experiments; *n* = 50,000 cells) of $P_{traG}$ promoter reporting system activating cell fluorescence (FITC-A) as determined by GFP fluorescent cells counted using flow cytometry (**D**) Flow cytometry histogram for $PtraG$-$sfGFP$ reporter signal from samples taken at OD0.5 and OD1.5 representing cell count in FITC-A expression level (experiment was repeated 3 times with similar results). Source data and detailed statistical results are displayed in Source Data file.

incubated at 30 °C and shaken every 20 min. Optical density (600 nm) was recorded after shaking using a Synergy H1 microplate reader (BioTek, Winooski, VT). For the competition experiment (4 biological replicates), overnight cultures were washed in LB medium and mixed (*Pseudomonas putida* KT2440 + *Pseudomonas putida* KT2440/pKJK5) equally. The mixes were diluted $10^4$ times, and 800 μL was distributed in a 12-well plate. Total cell number and plasmid carriers were counted before and after competition using a $10^5$-fold dilution plated on LB plates and LB and tetracycline 20 μg/mL, respectively. The fitness of the plasmid was calculated as $w = \ln(Pcarrier_f/Pcarrier_i)/\ln(Pfree_f/Pfree_i)$. For the transcriptomics analyses, 100 mL cultures were incubated in flasks from a 200-fold dilution and sampled at 600 nm optical densities of 0.5 and 1.5.

### GFP-based reporter plasmid and flow cytometry analysis

The promoter $PtraG$ was cloned using HiFi Gibson Assembly into a pPROBE-NT backbone with primers traGp_Fw (5′-gttagttagggaataagccgagttttaagggagcctcgcgg-3′) and traGp_Rv (5′-aggtcgactctagaggatcggccaggaagagggctaaag-3′) to amplify the $P_{traG}$ region from pKJK5 with overhangs to pPROBE-NT (Addgene Plasmid #37818) and primers MFHO38 (5′-gatcctctagagtcgacctgc-3) and MFHO37 (5′-tcggcttattccctaactaactaaag-3) for pPROBE-NT. The GFP reporter plasmids were transformed into *P. putida* KT2440 (negative control) and *P. putida* KT2440/pKJK5 following the electroporation procedure. Overnight cultures were diluted 1/50 and normalized to an OD of 0.1. These cultures (*n* = 3) were then grown at 30 °C and 250 RPM. For cell size and granulometry, samples were taken at OD0.5 and 1.5 and examined by flow cytometry (FACS Aria III, Becton Dickinson Biosciences, San Jose,

CA, USA) using the 488 nm laser and the FSC and SSC detection channels, respectively (*n* = 3 independent experiments). To measure GFP intensity and abundance, samples were taken at OD0.5 and OD1.5, washed twice with PBS and examined by flow cytometry using a 488 nm excitation laser and the FITC (530/30 nm bandpass filter) detector channel. The closing gate was set at 50,000 counts using the FSC and SSC detection channels. A culture of *P. putida* KT2440/pKJK5 containing the pPROBE-NT plasmid was used as a control to set up the gate for the GFP signal on the FITC detection channel. The data in Fig. 4 were analyzed using FlowJo software (Tree Star Inc., USA).

### Transcriptomic data acquisition

After an overnight culture of 16 h in 5 mL of selective LB (250 RPM), *Pseudomonas putida* KT2440 and *Pseudomonas putida* KT2440/pKJK5 were washed and diluted 500 times in 100 mL of LB. Flasks were incubated at 30 °C (250 RPM), and cultures were sampled at turbidity (600 nm) of 0.5 and 1.5.

**Population-level (bulk) RNA-seq.** Samples (*n* = 4 independent experiments) were centrifuged (3 min, 7000 × *g*), RNA content was directly extracted using the Quick-RNA Fungal/Bacterial Miniprep Kit (Zymo Research, CA, USA) following the manufacturer's instructions, and libraries were prepared using the Zymo-Seq RiboFree Total RNA Library Kit (Zymo Research, CA, USA). Fragment integrity and size were assessed using a fragment analyzer (Agilent, CA, USA) and quantified with a Qubit dsDNA HS assay kit (Thermo Fisher Scientific, Waltham, MA, USA) before sequencing on a Novaseq 6000 (Illumina, CA, USA) at 2 × 150 bp performed by Novogene Co., Ltd. (Cambridge,

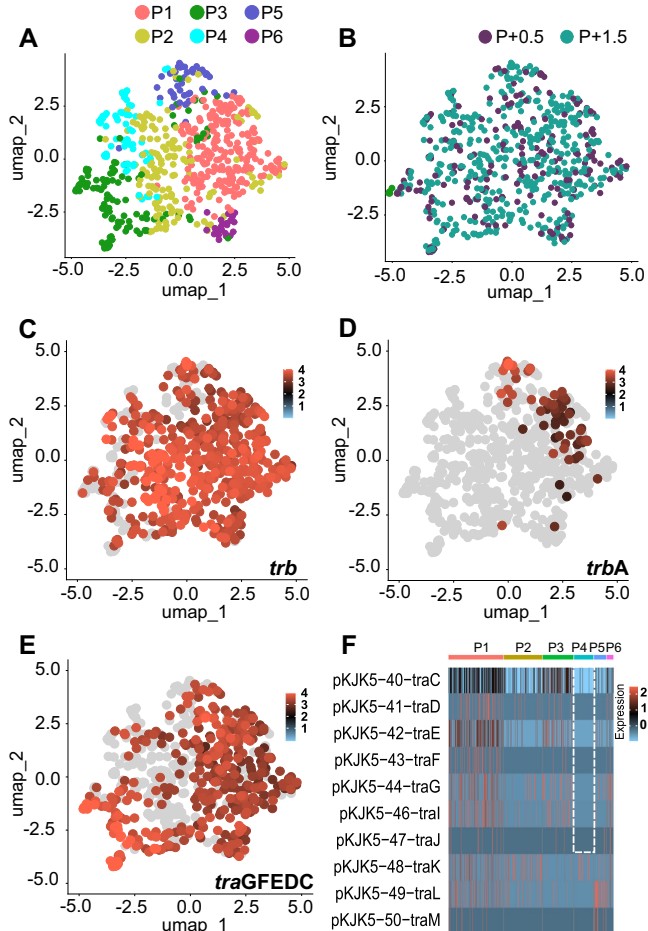

**Fig. 5 | Plasmid single-cell transcriptome clustering suggests population heterogeneity in plasmid gene transcription, with differential expression of *tra* genes.** Subpopulation clustering identified by single-cell transcriptomics of *P. putida* plasmid carriers at early (OD0.5) and late exponential growth (OD1.5) (*n* = 621 P+ cells, Table S3). **A** UMAP obtained by microSPLiT scRNA sequencing identified 6 plasmid transcriptome clusters (P1-P6) when only plasmid transcripts were analyzed with 6 dimensions (see Fig. S12). Clustering was performed with an integrative approach combining Euclidean distance-based K-nearest neighbor, refining the edge weights with the Jaccard similarity and the Louvain algorithm. Same UMAP, where dots were colored according to (**B**) the OD of the cell or (**C**–**E**) to normalized transcript number of (**C**) all *trb* genes, (**D**) *trb*A genes, (**E**) *tra*GFEDC genes. Normalization consist in a log10 transformation of transcript numbers+1 * 10,000 divided by summed plasmid transcript number of the cell. **F** Heatmap of the normalized centered-scaled number of *tra* transcripts per cell ordered according to clusters P1-6. Source data are provided as a Source Data file.

United Kingdom). Sequence files can be found at NCBI under Bioproject ID PRJNA1019643.

**microSPLiT scRNA-seq.** The experiment was repeated 2 times (*n* = 2 independent experiments) following the microSPLiT protocol[7]. Samples were centrifuged (3 min, 7000 × *g*), resuspended in the same volume of cold paraformaldehyde 4% (pH 7.2) for fixation and kept at 4 °C overnight. Fixed samples were permeabilized by lysozyme, and a polyA tail was added to mRNA using *E. coli* Poly(A) Polymerase (New England Biolabs, Ipswich, MA). Cell subsamples were stained with SYBR™ Green I (Thermo Fisher Scientific, Waltham, USA), and their concentration in each sample was measured using a FACS Aria III (Becton Dickinson Biosciences, San Jose, CA, USA). A total of 250,000 cells were distributed in 48 wells where reverse transcription occurred (Maxima H Minus Reverse Transcriptase, Thermo Scientific™, Thermo Fisher Scientific, Villebon sur Yvette. France), each well containing

primers with a specific barcode. Cell distribution was carried out blindly during the first experiment, while P-0.5, P-1.5, P + 0.5 and P + 1.5 samples were knowingly distributed among the 48 wells. After reverse transcription, the cells were pooled together and physically separated (vortex and filtration steps at 10 and 1 μm, pluriSelect Life Science UG (haftungsb.) & Co. KG, Leipzig, Germany) and randomly split into 96 wells where a well-specific oligo was added to the cDNA construct by ligation with a T4 DNA Ligase (New England Biolabs, Ipswich, MA). Cell pooling, random splitting and ligation were repeated for a third barcode addition so that, statistically, cell-hosted cDNAs carry a unique 3-barcode combination. Cells were pooled, divided into sublibraries of 10,000 (replicate 1) or 6000 cells (replicate 2) and lysed after measuring their concentration using a FACS Aria III (Becton Dickinson Biosciences, San Jose, CA, USA). The second strand synthesis was performed using template switching oligos, and cDNAs were amplified using a KAPA HiFi HotStart ReadyMix PCR kit (Roche Sequencing Solutions, Pleasanton, CA) supplemented with EvaGreen® Dye (Biotium, San Francisco, CA) for 14 cycles in total. Sublibraries were fragmented using Enzymatics Fragmentase (QIAGEN Beverly, Beverly, CA) and amplified. Sublibrary-specific adapters were ligated using Enzymatics ligase (QIAGEN Beverly, Beverly, CA), and sublibraries of cDNA were amplified using a KAPA HiFi HotStart ReadyMix PCR kit (Roche Sequencing Solutions, Pleasanton, CA).

Fragment integrity and size were assessed using a Fragment Analyzer (Agilent, CA, USA) using an NGS Fragment Kit (1–6000 bp) and quantified with a Qubit dsDNA HS assay kit (Thermo Fisher Scientific, Waltham, MA, USA) before sequencing on a Novaseq 6000 (Illumina, CA, USA) at 2 × 150 bp performed by Novogene Co., Ltd. (Cambridge, United Kingdom). A list of primers and associated barcodes can be found in Kuchina et al. [7]. Sequence files can be found at NCBI under Bioproject ID: PRJNA1019643. The number of cells and transcripts per cell were equivalent in the different generated sublibraries (Fig. S4B).

## Computational method

The reference genome was generated using STAR 2.7.9a (https://github.com/alexdobin/STAR) combining the pKJK5 plasmid and *Pseudomonas putida* KT2440 genome using, respectively, GenBank record AM261282.1 and assembly GCA_000007565 from EnsemblBacteria. The reference genome was indexed for STAR using parameters (--genomeSAindexNbases 10) specific for small genomes using the formula min(14, log2 (Genome Length)/2 - 1). The created genome was used as a reference for both approaches (bulk & microSPLiT).

**Population-level (bulk) RNA-seq.** The sequenced reads were trimmed of the remaining adapter sequences and low quality based using bbduk (BBMap 38.90–Bushnell B. – sourceforge.net/projects/bbmap/), with right-side trimming using parameters {*k* = 23 mink=11 hdist=1 tpe tbo trimq=10}. Trimmed reads were mapped against the reference genome, and the per-cell gene counts were quantified using STAR 2.7.9a (https://github.com/alexdobin/STAR). Transcripts identified as rRNA, tRNA and plasmid encoded were removed from the data. For heatmaps, feature counts from the generated contingency table (gene × sample) were normalized (divided by the total counts, multiplied by 10,000 + 1, and log10 transformed) and centered-scaled with the R scale function.

**microSPLiT scRNA-seq.** Sequenced reads were demultiplexed (barcode list can be found in ref. 7) and mapped against the reference genome, and the per-cell gene expression was quantified using STARsolo (STAR 2.7.9a; https://github.com/alexdobin/STAR/). A matrix of unique molecular identifier (UMIs, unique gene transcript-associated barcode) counts for each cell (N-by-K matrix, with N cells and K genes) was generated by gathering cells from all sublibraries. UMIs identified as rRNA or tRNA were removed from the dataset, and cells with less than 85 UMIs were sorted out. This threshold was chosen

to be above the sharp drop of UMIs[48] as determined with the knee-graph (UMI count per barcode rank; Fig. S3). This left samples sizes of 3165 cells (Control E1 experiment) and 1486 cells (E2 experiment). We consider that this sample size was sufficient since cell identity could be retraced from the clustering[48]. UMIs identified as plasmid genes were counted and added as metadata to characterize cells, as well as the sublibrary, the associated experiment and the sample type. For each subset of data (whole-genome encoded genes; chromosome encoded genes or plasmid encoded genes), a SeuratObject was created using the R package Seurat 5.0.1. (https://cran.r-project.org/web/packages/Seurat/index.html) with R 4.3.2. Genes presents in less than 10% of cells in all cell types (P-0.5; P-1.5; P + 0.5; P + 1.5) were sorted out for subsets of data made of whole-genome encoded genes and chromosome encoded genes in order to ensure a robust analysis[49]. Feature counts per cell were divided by the total counts of the cell, normalized (divided by the total counts, multiplied by 10,000 + 1, and log10 transformed), scaled and centered using the scale R function (heatmaps) or linear model with the ScaleData function from Seurat R package for cell clustering. Variables were individually regressed against each feature, and the resulting residuals are then scaled and centered. Linear dimensional reduction was performed with a PCA, and clustering parameters were selected by combining a JackStraw resampling test and an elbow plot (Fig. S7). Cells were clustered with an integrative approach combining Euclidean distance-based K-nearest neighbor, refining the edge weights with the Jaccard similarity and the Louvain algorithm. A nonlinear dimensionality reduction was then applied with UMAP. Inter-gene Spearman correlation and associated $p$ values were calculated with the *rcorr-5.1-2* R package (https://search.r-project.org/CRAN/refmans/Hmisc/html/rcorr.html).

## Reporting summary

Further information on research design is available in the Nature Portfolio Reporting Summary linked to this article.

## Data availability

The sequence data generated in this study have been deposited in the National Center for Biotechnology Information's SRA repository under accession code PRJNA1019643. The reference genome data used in this study were generated combining the pKJK5 plasmid sequence available in the NCBI database under accession code AM261282.1 and *Pseudomonas putida* KT2440 genome available in the Ensembl Bacteria database under accession code GCA_000007565. The mapping, demultiplexing and quantification for bulk and single cell RNA-seq data generated in this study have been deposited in the Zenodo database under accession code 11356666. Source data are provided with this paper.

## Code availability

Statistical analyses were performed with R 4.3.2 (packages vegan-2.6-4, mvabund-4.2.1, Seurat-5.0.1, patchwork-1.2.0, dplyr-2.4.0, scCustomize-2.0.1, ggplot2-3.4.4, rcorr -5.1-2 and corrplot-0.92). The sequenced reads were trimmed of the remaining adapter sequences and low quality based using bbduk (BBMap 38-90-Bushnell B. - sourceforge.net/projects/bbmap/). Reads were mapped against the reference genome, and the per-cell gene counts were quantified using STAR-2.7.9a (bulk trascriptomics) or STARsolo from STAR-2.7.9a (single-cell transcriptomics) (https://github.com/alexdobin/STAR). Flow Cytometry analyses were obtained with FlowJo software v10 (Tree Star Inc., USA).

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

## Acknowledgements

This work was supported by Villum Fonden (BioRep-HGT, Grant No. 00028304) and by the Fond National de Recherche Scientifique (F.R.S-FNRS). A.K. and G.S. acknowledge support from the Department of Energy Office of Science, Biological and Environmental Research (BER) Program, Grant DE-SC0023091. We acknowledge the use of computing resources at the core facility for biocomputing at the Department of Biology, University of Copenhagen. We would like to thank Prof. Lars Hestbjerg Hansen and Assistant Prof. Witold Piotr Kot for help in sequencing to verify the quality of the initial microSPLiT-seq libraries.

## Author contributions

V.C. and J.S.-M. conceived the study and designed the laboratory experiments. V.C. and R.I.-C. carried out the laboratory experiments and conducted the analysis of the experimental data. J.S.-M. and J.N. contributed to materials and analysis tools. A.K. and G.S. provided protocols and expertise. All authors discussed the theory and wrote the manuscript.

## Competing interests

G.S. is a co-founder and shareholder of Parse Biosciences, a scRNA-seq company. The rest of the authors declare no competing interests.
