## [Peer Review File · Nature Communications]

Single-cell RNA sequencing reveals plasmid constrains bacterial population heterogeneity and identifies a non-conjugating subpopulationEditorial Note: This manuscript has been previously reviewed at another journal that is not operating a transparent peer review scheme. This document only contains reviewer comments and rebuttal letters for versions considered at *Nature Communications*.

Reviewer #1 (Remarks to the Author):

I reviewed the previous version of this manuscript, and I am pleased to see that the authors have fully addressed my concerns and questions in this revision. I have no additional major comments or concerns and appreciate the additional analysis (Fig. 3A) and robustness checks done by the authors.

I believe this is a very interesting dataset, that would make for a valuable addition to the literature on phenotypic variation and plasmid biology. The conclusions are well supported by the data and the methods are now clearly explained.

Although the manuscript is generally well written and easy to follow, I did struggle a bit to quickly grasp some parts of the texts—requiring me to read several paragraphs multiple times—despite already having read them in detail during the initial review process. I believe some minor changes in the presentation could help make these paragraphs easier to follow for first time readers (see specific suggestions below).

In general, I found it a bit hard that many important figures are currently in the Supplement: to fully grasp the text I had to go back and forth between text, main figures, and supplement, and this made it harder to maintain an overview. If the guidelines allow for it—and the editor agrees—I would thus suggest to move some (parts of) the more essential SI figures (e.g. Fig S9 / S10 / S12D / S14) to the main text.

Minor suggestions for clarification:

Lines 152-161: I had to read this multiple times before fully grasping this, one reason is that the cluster labeling makes it hard to connect the figure to text vice-versa: I had to go back multiple times to remind myself that W1, W4, W7, W10, W11 belong together and W2, W3, W5, W6, W8, W9 belong together. It would be good to help the reader connect things more quickly. One option might be to relabel clusters in a more organized way (e.g. top-to bottom, column wise) or in another way visually connect things better (e.g. in text you can explicitly say left and right part, or positive/negative umap 1 to group these clusters).

Lines 173-176: related to comment above, it would help to connect this better to figure by explicitly referring to cluster number in text.

Lines 180: I'm a bit confused by this: does the low growth rate refer to the measured growth curves (Fig 1A) or is this inferred from cluster analysis of gene transcripts. The "hence" at the start of the sentence suggests the latter, but this does not connect well with discussion at the end of the section above. I suggest to rephrase this sentence.

Line 181-183: I do not really understand this sentence. There seems to be a word missing after "between". Also, it is not directly obvious what "gene cluster coefficient" means. I think this needs a bit more explanation.

Line 184 "unsupervised cluster analysis plasmid genes" missing "on" after analysis?

Line 179-192: Generally, I found this paragraph hard to read. It is quite dense in technical details and it is not always fully clear where the authors are going with this. I would suggest to help the reader a little more by explaining in some more detail what is being done and what this means.

Line 208: missing S in figure nr.

Line 210: idem.

Fig S12D: mistake in caption? Labels in Figure (W cluster labels) do not match caption (red / blue plasmid +/- labels). Also the colored points are too small to see well, suggest to make them bigger.

Reviewer #3 (Remarks to the Author):

The authors have incorporated my suggestions with text revisions.

University of Copenhagen, Denmark
May 10th, 2024

Dear referees,

We would like to thank you for reviewing the manuscript entitled: “Single-cell RNA sequencing reveals plasmid constrains bacterial population heterogeneity and identifies a non-conjugating subpopulation” a second time and for providing positive and valuable feedbacks on the new manuscript.

You will find below, a response to the different comments (in a blue font), indicating what was changed in the last manuscript to address these comments. We appreciate the thoroughness and insightfulness of your comments, which have undoubtedly strengthened the quality of our work.

Best regards,

Valentine Cyriaque, Jonas S. Madsen, and co-authors

Reviewer #1 (Remarks to the Author):

I reviewed the previous version of this manuscript, and I am pleased to see that the authors have fully addressed my concerns and questions in this revision. I have no additional major comments or concerns and appreciate the additional analysis (Fig. 3A) and robustness checks done by the authors.

I believe this is a very interesting dataset, that would make for a valuable addition to the literature on phenotypic variation and plasmid biology. The conclusions are well supported by the data and the methods are now clearly explained.

Although the manuscript is generally well written and easy to follow, I did struggle a bit to quickly grasp some parts of the texts—requiring me to read several paragraphs multiple times—despite already having read them in detail during the initial review process. I believe some minor changes in the presentation could help make these paragraphs easier to follow for first time readers (see specific suggestions below).

In general, I found it a bit hard that many important figures are currently in the Supplement: to fully grasp the text I had to go back and forth between text, main figures, and supplement, and this made it harder to maintain an overview. If the guidelines allow for it—and the editor agrees—I would thus suggest moving some (parts of) the more essential SI figures (e.g. Fig S9 / S10 / S12D / S14) to the main text.

R: We greatly appreciate the feedback and added more information in the text for a better understanding (see next comments). We also moved the Figures S9, S10 and S14 to the main text (now figure 2 and 5)

Minor suggestions for clarification:

Lines 152-161: I had to read this multiple times before fully grasping this, one reason is that the cluster labeling makes it hard to connect the figure to text vice-versa: I had go back multiple times to remind myself that W1, W4, W7, W10, W11 belong together and W2, W3, W5, W6, W8, W9 belong together. It would be good to help the reader connect things more quickly. One option might be to relabel clusters in a more organized way (e.g. top-to bottom, column wise) or in another way visually connect things better (e.g. in text you can explicitly say left and right part, or positive/negative umap 1 to group these clusters).

A clear separation was added on the figure (Fig. 3) and names were attributed to the group of subpopulations. In the text, we detailed the subpopulations included in the 2 different groups and refer to them later with these names.

Lines 173-176: related to comment above, it would help to connect this better to figure by explicitly referring to cluster number in text.

Cluster numbers and indication on the heatmap were added to the text (now lines 175-178)

Lines 180: I'm a bit confused by this: does the low growth rate refer to the measured growth curves (Fig 1A) or is this inferred from cluster analysis of gene transcripts. The "hence" at the start of the sentence suggests the latter, but this not connect well with discussion at the end of the section above. I suggest rephrasing this sentence.

"Hence" was removed from the sentence so that it is now clearer that low growth rate refer to the measured growth curves and "(Fig 1A)" was added to the end of the sentence (lines 180-182).

Line 181-183: I do not really understand this sentence. There seems to be a word missing after "between". Also, it is not directly obvious what "gene cluster coefficient" means. I think this needs a bit more explanation.

The sentence was rephrased as "The burden of the plasmid may also have resulted in the observed decrease of the cluster coefficient in plasmid carrier cells (degree to which genes tend to cluster together ; Tab. S6), in network graphs calculated from spearman correlations between genes of each single-cell at OD0.5 (>85 transcripts/cell ; Fig. S13). Reduced co-occurrence correlation between chromosomal genes of plasmid carrier cells suggests that the plasmid interferes with chromosome transcriptional regulation." where the gene cluster coefficient was defined (line 182-186).

Line 184 "unsupervised cluster analysis plasmid genes" missing "on" after analysis?

The sentence was reformulated as "Furthermore, when calculating the Spearman correlation between plasmid and chromosomal genes (Fig. 3A), we see that plasmid

encoded genes, especially *tra* genes, were transcribed concomitantly ($n > 30$, $p < 0.05$).” (lines 187-189).

Line 179-192: Generally, I found this paragraph hard to read. It is quite dense in technical details and it is not always fully clear where the authors are going with this. I would suggest to help the reader a little more by explaining in some more detail what is being done and what this means.

As detailed in the responses of the other comments, changes were made in the text to clarify the point.

Line 208: missing S in figure nr.

Line 210: idem.

We apologize for these mistakes : this figure is now Fig.5.

Fig S12D: mistake in caption? Labels in Figure (W cluster labels) do not match caption (red / blue plasmid +/- labels). Also the colored points are too small to see well, I suggest to make them bigger.

Caption of figure S12D was mistakenly associated with the end of caption of figure S13. This lines were then moved to the caption of figure S13 and replaced by the appropriate caption in figure S12, now recalled Fig. S10.

Dots were drawn bigger.